# Polydispersity vs. Monodispersity. How the Properties of Ni-Ag Core-Shell Nanoparticles Affect the Conductivity of Ink Coatings

**DOI:** 10.3390/ma14092304

**Published:** 2021-04-29

**Authors:** Anna Pajor-Świerzy, Dawid Staśko, Radosław Pawłowski, Grzegorz Mordarski, Alexander Kamyshny, Krzysztof Szczepanowicz

**Affiliations:** 1Jerzy Haber Institute of Catalysis and Surface Chemistry Polish Academy of Sciences, Niezapominajek 8, 30239 Kraków, Poland; stasko.d@outlook.com (D.S.); nbmordar@cyf-kr.edu.pl (G.M.); ncszczep@cyf-kr.edu.pl (K.S.); 2Abraxas Jeremiasz Olgierd, Piaskowa 27, 44300 Wodzisław Śląski, Poland; radek.pawlowski@helioenergia.com; 3Casali Center for Applied Chemistry, Institute of Chemistry, Edmond J. Safra Campus, The Hebrew University of Jerusalem, Jerusalem 91904, Israel; alexander.kamyshny@mail.huji.ac.il

**Keywords:** core-shell nanoparticles, monodispersity and polydispersity, thermal sintering, conductive properties

## Abstract

The effect of polydispersity of nickel-silver core-shell nanoparticles (Ni-Ag NPs) on the conductivity of ink coatings was studied. Ni-Ag NPs of various average diameters (100, 220, and 420 nm) were synthesized and utilized for the preparation of conductive inks composed of monodisperse NPs and their polydisperse mixtures. The shell thickness of synthesized Ni-Ag NPs was found to be in the range of 10–20 nm and to provide stability of a core metal to oxidation for at least 6 months. The conductivity of metallic films formed by inks with monodisperse Ni-Ag NPs was compared with those formed by polydisperse inks. In all cases, the optimal conditions for the formation of conductive patterns (weight ratio of monodisperse NPs for polydisperse composition, the concentration of the wetting agent, sintering temperature, and duration) were determined. It was found that metallic films formed by polydisperse ink containing 100, 220, and 420 nm Ni-Ag NPs with a mass ratio of 1:1.5:0.5, respectively, are characterized by the lowest resistivity, 10.9 µΩ·cm, after their thermal post-coating sintering at 300 °C for 30 min that is only 1.6 higher than that of bulk nickel.

## 1. Introduction

During the last years, nanoscience and nanotechnology have attracted much attention due to their great impact on the fabrication of functional nanomaterials with novel physical and chemical properties. Among nanosized particles, metal nanoparticles (NPs) are nowadays widely applied for the preparation of conductive coatings for electronic circuits and devices such as batteries, electro-optic devices, Radio Frequency Identification (RFID) tags, thin-film transistors, transparent conductive electrodes, touch panels, and flexible displays [1,2,3,4,5,6]. Various methods of deposition of metal nanoparticles dispersed in a liquid vehicle, such as spin, spray, dip and rod coatings, screen, inkjet, flexo, and gravure printings are used for the fabrication of conductive films [6,7].

The conductivity of such coatings strongly depends on the type of metal and the tightness of nanoparticles packing in the deposited film [8,9]. Such tight packing is reached by post-coating sintering, which results in the formation of numerous contacts and percolation paths between metal NPs due to removal of organic components of inks (e.g., polymeric stabilizers, wetting agents) and even welding of NPs. Such effect can be achieved by heating (thermal sintering), the action of intense light (photonic sintering), plasma, microwave, and chemical sintering [6,10].

As to the type of nanoparticles, currently, most conductive nanoinks are based on silver NPs. Silver is a noble metal, which is resistant to oxidation and possesses the highest electrical conductivity among metals. However, large-scale production of printed electronic devices requires low-cost nanoinks, in which silver as a conductive functional material should be replaced by lower-cost metals with high electrical conductivity, for example, copper and nickel. However, the specific challenge while utilizing NPs of such metals is their oxidation at ambient conditions. To overcome this disadvantage, NPs of these metals should be protected against oxidation, and the most effective approach toward obtaining stable metal NPs applicable for conductive ink formulations is the formation of a dense shell composed of a protective non-oxidizable highly conductive material such as noble metals, i.e., formation of bimetallic core-shell NPs. Such hybrid NPs can be obtained by transmetalation, which is a galvanic displacement reaction when the surface of preformed core functions (and is sacrificed) as a reducing agent for the second metal with higher reduction potential. By this process, the reduction of the second metal occurs only on the surface of the core metal, resulting in the formation of a metal shell on the surface of the core metal [6,11].

It can be noticed that most of the inks for the preparation of conductive coatings are based on metal NPs with monodisperse size distribution [12,13]. Although monodisperse NPs provide better stability of ink formulations, which is especially important for optimal printing performance (avoiding aggregation and precipitation of NPs and nozzle clogging) [14], some specific requirements have to be addressed and several specific challenges should be overcome for the obtaining nanoparticles with monodisperse size distribution. To improve the monodispersity and prevent aggregation of metal NPs, the mass ratio of the stabilizers-to-precursors in the synthesis process should be usually high [15], which not only increases the cost of the fabrication of electronic circuits but can also decrease their conductivity due to the presence of a large amount of organic insulating material. In addition, it is also quite challenging to prepare the dispersions of monodisperse NPs of high concentrations (high metal loading is required for obtaining highly conductive patterns) and on a large scale. Considering the geometric aspects of coating, more tight packing of NPs can be obtained while using polydisperse nanoparticles (voids between large NPs are filled with small NPs), and hence higher conductivity can be achieved. In this context, we should ask, is it rational to pay so much effort to obtain perfect monodisperse systems? Is it crucial the obtaining metal NPs with monodisperse size distribution for the preparation of highly conductive ink coatings?

It was found that the conductivity of nanocomposite materials can be improved by using NPs with bimodal size distribution. For example, Balantrapu et al. [16] obtained the lowest resistivity in the sintering temperature range of 125–200 °C for the ink coating composed of a mixture of silver NPs with sizes 12 and 80 nm. The enhancement of conductivity was also observed for the metallic film containing a mixture of Ag NPs with sizes 9 and 170 nm as compared with the one based on NPs with monodisperse size distribution [17]. In our recent work, we found that the ink coatings composed of Ni-Ag NPs with bimodal sizes distribution (70 and 250 nm) had higher conductivity than the films based on monomodal NPs [9].

In the present research, we focused on the effect of polydispersity of Ni-Ag NPs on the final conductive properties of metallic coatings. Polydisperse dispersions of Ni-Ag NPs for ink formulation were prepared by mixing of three types of Ni-Ag NPs with different average sizes synthesized by previously developed methods [9,18,19] with some additional improvements. The conductive properties of ink coatings composed of monodisperse and polydisperse Ni-Ag NPs were analyzed and compared.

## 2. Materials and Methods

### 2.1. Materials

Nickel sulfate hexahydrate (NiSO_4_·6H_2_O), sodium borohydride (NaBH_4_), sodium carboxymethyl cellulose (CMC) with MW 90000, and amino methyl propanol (AMP) were purchased from Sigma-Aldrich (Poznań, Poland). Citric acid (CA) and silver nitrate (AgNO_3_) were products of Avantor Performance Materials Poland S.A (Gliwice, Poland). Wetting agent (TEGO WET KL 245, polyether siloxane copolymer), was obtained from Evonik (Essen, Germany).

### 2.2. Synthesis of Ni-Ag NPs

To obtain Ni-Ag NPs with different average sizes, their synthesis was performed by using an improved two-stage process recently developed in our lab [9,18,19], in the presence of various complexing agents. At the first stage, Ni NPs were synthesized by the “wet” chemical process, while at the second stage, the Ag shell was formed on the surface of Ni NPs by the transmetalation reaction.

Ni NPs were obtained by the reduction of Ni ions in complex with CA and AMP, in the presence of sodium CMC (MW = 90,000) as a stabilizer. In the optimized process, 30 mL of an aqueous solution of CMC (0.5%) and 12 mL of the NiSO_4_ aqueous solution (0.2 M) were mixed and then 12 mL of the aqueous solution of CA (complexing agent) was added to obtain its final concentration 0.15 M. Thereafter, AMP solution with concentration 80% was added to the above mixture to adjust the required pH value in the range of 6.5–12. Finally, 30 mL of a reducing agent, NaBH_4_, a solution with a concentration of 0.05 M was injected into the reaction mixture. The mixture was stirred at 850 rpm for 60 min. For obtaining Ni NPs with different average sizes, the different amounts of AMP and the presence/absence of CA in the reaction mixture were used. In addition, the properties of Ni NPs obtained in the presence of AMP were compared with the properties of Ni NPs synthesized in the reaction mixtures of the same pH range, but adjusted with 0.1M NaOH.

At the next stage of Ni-Ag NPs synthesis, to the obtained dispersion of Ni NPs, 45 mL of the aqueous dispersion of the precursor of a silver shell (AgNO_3_), with concentrations in the range of 0.01–0.08 M was added. The synthesis was performed for 60 min at room temperature while stirring at 850 rpm.

### 2.3. Ink Composition and Conductive Coatings Fabrication

The synthesized Ni-Ag NPs were washed two times with distilled water and concentrated to 25 wt% by centrifugation (7000 rpm, 20 min). Then, dispersions of these NPs with various average sizes as well as their mixtures with various weight ratios (polydisperse dispersions) were used for ink preparation. The physicochemical properties of NPs based inks were optimized by using TEGO WET KL 245 (Evonik, Essen, Germany) as a wetting agent at various concentrations. The obtained inks were homogenized by using an ultrasonic bath for 30 min at 20 kHz. The ink coatings were formed on glass slides by using a bar coating method (with 0.05 mm of wire winding rod) [20]. After the deposition process, the ink layer was dried on a hot plate at 60 °C for 10 min and then the obtained films were sintered by heating at various temperatures (250–350 °C) during 15–90 min.

### 2.4. Characterization

The hydrodynamic diameter of nanoparticles according to number distribution was measured by the dynamic light scattering (DLS), and the zeta potential by the microelectrophoretic method using a Zetasizer Nano Series system (Malvern Instruments, Malvern, Worcestershire, UK). Each value was determined as an average of three runs with at least 20 measurements. All analyses were performed with aqueous dispersions of NPs at 25 °C. The optical properties of the dispersions of Ni-Ag NPs were studied by UV-Vis spectrophotometry (UV-1800, Shimadzu, Kyoto, Japan). The chemical composition of Ni-Ag NPs was evaluated by using freshly synthesized dispersion of NPs dried in vacuum at room temperature by the use of X-ray photoelectron spectroscopy (XPS) with an ESCA/XPS equipped with a semispherical analyzer EA15 (Prevac, Rogów, Poland) using Al-Kα (1486.6 eV) radiation with a power of 180 W. For the Ag 3d5/2 line, the resolution of the instrument was 0.9 eV, and the spectra were acquired at a pressure of 2 × 10^−9^ mbar. The efficiency of the reduction process of Ni ions was evaluated by the Energy Dispersive X-ray fluorescence technique (EDXRF, FISCHERSCOPE X-RAY XDL 230, Sindelfingen, Germany). To establish the concentration of Ni NPs, the calibration curve was defined by the measurement of the intensity of X-ray fluorescence at various concentrations of Ni ions.

The ink formulation was deposited by using a hand coater (Kontech, Łódź, Poland). The morphology (shape, size) of synthesized NPs and their coatings were visualized by SEM (LEO Gemini 1530, Zeiss, Jena, Germany).

The thickness of coatings was measured by the EDXRF method (FISCHERSCOPE X-RAY XDL 230). The sintering process was performed on a hot plate in an atmospheric environment. The sheet resistance of metallic coatings was determined by a four-point probe method (Milliohm Meter, Extech Instruments, Nashua, NH, USA). The values of the resistivity were calculated by multiplying the values of the sheet resistance by the thickness, and the conductivity was established as the reciprocal value to the resistivity [21].

## 3. Results and Discussion

Taking into account the geometrical aspects, it seems to be rational, that the polydispersity of metallic NPs will favor the formation of films with higher conductivity as compared with monodisperse NPs, since smaller NPs can occupy voids between larger NPs that will lead to the formation of a metallic layer with tightly packed structure. To confirm experimentally this assumption, the coatings formed by monodisperse Ni-Ag NPs of various sizes and by their mixture were fabricated in a multistep process and their conductivities were compared. The scheme of preparation of coatings composed of Ni-Ag NPs of different sizes is shown in Figure 1.

### 3.1. Preparation of Polydisperse Ni-Ag NPs Dispersion

To prepare polydisperse Ni-Ag NPs dispersion, the previously developed methods of synthesis Ni-Ag NPs were adopted with some modification [18,19]. These two-step methods consist of a synthesis of Ni NPs followed by silver shell formation by transmetalation reaction [11]. At the first step, NiSO_4_ (0.2M) was used as a precursor of Ni NPs, while NaBH_4_ as a reducing agent, the reduction was performed in sodium carboxymethyl cellulose solution (0.5%) as a stabilizer. The synthesis of Ni NPs was carried out at the deficiency (0.05M) of the reducing agent that is important for the stage of the formation of a silver shell by transmetalation [11,22]. Due to its deficiency, all NaBH_4_ is consumed in the reduction of Ni ions that prevents the reduction of Ag ions to Ag NPs instead of reduction of Ag ions on the surface of Ni NPs with the formation of the silver shell. As a result of this process, the Ni NPs with an average size of about 250 nm were obtained (Appendix A, Appendix A). It is a well-known phenomenon that at the alkaline pH the redox potential of sodium borohydride increases, which favors the reduction of metal ions [23]. Therefore, the reduction of Ni ions was performed in an alkaline medium. It was achieved by the addition of AMP or NaOH. The addition of both reagents to only slightly alkaline pH~8 results in precipitation of green gelatinous Ni(OH)_2_. To avoid its formation, citric acid (CA) as a complexing agent of Ni ions was used. Such an approach was also reported by others [24,25,26]. In the present research, CA with a concentration of 0.15M was used that is below the stoichiometric balance with Ni ions, which provides its total consumption by Ni ions with the formation of Ni-CA complex. Again, this is important for the formation of a silver shell by transmetalation, since CA is also a good reducing agent for Ag ions, and its excess in the reaction mixture can lead to the formation of Ag NPs along with the formation of Ag shells. The decreasing of the size of Ni NPs from about 250 to 220 nm (Appendix A, Appendix A) in the presence of citric acid (without AMP and NaOH) may indicate the process of Ni-CA complex formation.

The efficiency of the reduction of Ni ions to Ni NPs for various compositions and pH was studied by the EDXRF method, and the obtained results are presented in Figure 2.

As seen from Figure 2, the amount of reduced Ni ions and, accordingly, the amount of formed metallic Ni increase from 13% to 91% and from 11% to 28% in the presence of AMP and NaOH, respectively with increasing pH from 6.5 to 12.0. This data testifies also to much higher efficiency of Ni ions reduction in the presence of AMP as compared with NaOH. The greater impact of AMP in comparison with NaOH can be explained by the formation of mixed Ni complex with AMP, which contains two functional complexing groups, amine and hydroxyl, and with CA (AMP-Ni-CA). It is worth noting that in the case of AMP excess (pH~12) without CA, the dissolution of Ni(OH)_2_ occurs that follows by changing the color of the Ni^2+^ solution to dark blue which indicates the formation of Ni-AMP complex. At excess of AMP, the efficiency of the reduction process was slightly lower (78%, red bar) than in the presence of both complexing agents (91%, black bar) that can also testify to the formation of the mixed AMP-Ni-CA complex.

Three types of Ni NPs: synthesized at pH 9.5 with CA and AMP, at pH 12 with CA and AMP, and at pH 12 with an excess of AMP were characterized by the size and polydispersity index. As seen from the data presented in Table 1, the size and polydispersity strongly depend on the conditions of the synthesis.

In the presence of both complexing agents, AMP and CA, much smaller NPs with average sizes of 80 and 180 nm are formed at pH 12 and 9.5, respectively, in comparison with 400 nm in the presence of only AMP as a complexing agent. More data on the effect of complexing agents on Ni NPs size distribution is presented in the Appendix A (Appendix A).

As was found, the synthesized Ni NPs are not stable and totally oxidize in aqueous dispersion after 4–5 h of storage. To stabilize Ni NPs against oxidation, a protective silver shell was formed on their surface by transmetalation reaction, which consists of the reduction of silver ions on the surface of obtained Ni NPs, which play the role of a reducing agent. As was reported previously [18], to prepare a stable protective silver shell on Ni NPs, the transmetalation process has to be optimized. The silver film should be as thin as possible in order to decrease the amount of expensive silver, but thick and dense enough to ensure the long-term stability of Ni-Ag NPs to oxidation at ambient conditions.

First, we studied the effect of AgNO_3_ concentration in the range of 0.0–0.08 M, added to the dispersions of synthesized Ni NPs with different average sizes by measuring the size of obtained Ni-Ag NPs (Figure 3 and Figure 4A). The obtained results indicate that the thickness of the silver shell can be controlled by the concentration of Ag ions in the reaction, and the diameter of all NPs increases by the value of ~68 nm after the formation of the silver shell at the highest AgNO_3_ concentration. We selected the lowest AgNO_3_ concentration, at which there were the lowest indications of oxidation of formed Ni-Ag NPs. Such AgNO_3_ concentrations were found to be 0.01, 0.04, and 0.01 M for optimal coating of Ni NPs with average sizes 80, 180, and 400 nm, respectively (Appendix A, Appendix A). The average sizes of obtained Ni-Ag NPs at these AgNO_3_ concentrations were found to be 100, 220, 420 nm, respectively (in the text below these NPs are designated as 100-Ni-Ag NPs, 220-Ni-Ag NPs, and 420-Ni-Ag NPs). Therefore, the shell thicknesses for all synthesized core-shell NPs are in the range of 10–20 nm. SEM images (Figure 5) of core-shell NPs testify to their more or less spherical morphology, and their average sizes correlate well with the sizes measured by DLS.

Figure 6 presents the UV-visible spectra of various NPs. As seen, the spectra of Ni-Ag core-shell NPs are characterized by distinct peaks at about 420 nm typical for surface plasmons of nanosized silver (in forms of spheres, islands, plates, rings, and so on) [27] while bare Ni NPs have no peaks in the spectrum. The intensity of plasmon peaks increases with an increase of Ag shell thickness.

The chemical composition of synthesized NPs was evaluated by XPS analysis. Table 2 supports the XPS spectra (presented in Figure 7 and Figure 8) and shows the measured atomic percentages of metallic Ni and Ag as well as their oxides.

Figure 7 and Figure 8 show the Ni2p and Ag3d XPS spectra, respectively, obtained for Ni-Ag NPs with different sizes: (A) 100 nm, (B) 220 nm, (C) 420 nm. The spectra are similar for all samples and display two main component peaks regarding nickel (Figure 7) as well as silver (Figure 8) analysis. At binding energies of about 853 and 860 eV the characteristic peaks of metallic nickel and nickel oxide, respectively were observed in Ni2p_3/2_. In Figure 8, for Ag3d_5/2_ at binding energies of about 368 and 367 eV, the peaks of metallic silver and AgO, respectively, were detected.

It can be seen, that the atomic percentages of metallic Ni in the dispersions of Ni-Ag NPs with average sizes 100 and 420 nm are comparable to previously published values [9,18] and are higher than 50%, while for the dispersion of NPs with the size of 220 nm it is much higher, 84%, which can be the result of the different conditions of synthesis process (pH 12 vs. 9) Miyakawa et al. [28] noticed that pH of the dispersion of metallic nanoparticles is an important factor for preferential Ag deposition on the NPs core, and the most favored Ag shell formation on the Cu core was observed at pH 9.

The high content of Ni oxide presented in Table 2 and XPS spectrum (Figure 7) may be explained as follows. NiO is formed on the surface of Ni NPs at the stages of their synthesis and washing under ambient conditions. Therefore, the surface of such NPs only partially (if not at all) undergoes the transmetalation reaction. These NPs are a small portion of a total amount of Ni NPs, but only they determine the Ni and Ni oxide contents measured by XPS since for Al Kα excitation, the information on chemical composition can be obtained at the sample depth <10 nm, [29] and Ni and NiO cannot be identified in Ni-Ag NPs with Ag shell thickness of 10–20 nm. Most Ni NPs do not undergo oxidation, or oxidation is negligible during pre-transmetalation stages, and a dense Ag shell is easily formed on their surface. This explanation correlates well with the fact that after six months of storage at ambient conditions, the atomic percentage of NiO detected by the XPS method for all types of Ni-Ag NPs does not change, and more important that there are no indications of their oxidation in dispersions and inks as well indications of conductivity loss of coatings (see below) during the same period of storage (the uncoated Ni NPs totally oxidized after 4–5 h of the dispersion storage). These findings clearly indicate the long-term stability of Ni NPs coated by Ag shell to oxidation at ambient conditions.

### 3.2. Ink Composition and Fabrication of Conductive Coatings

To evaluate the size distribution of NPs in mixed dispersion, the model dispersion containing Ni-Ag NPs with average sizes 100, 220, and 420 nm and weight ratio 1:1:1 was prepared. The result is presented in Figure 4B. As clearly seen, this dispersion is characterized by high polydispersity, (PDI 0.560 ± 0.050), as compared with monosized dispersions (Figure 4A).

To optimize the composition of Ni-Ag NPs ink to be used for conductive coatings, four types of dispersions with various contents of different Ni-Ag NPs were prepared and concentrated to 25 wt% by washing/centrifugation. Then TEGO WET KL 245 as a wetting agent was added at various concentrations as described in our previous reports [9,19]. After homogenization by ultrasound, the inks were deposited on glass slides by a bar coating method (with 0.05 mm of wire winding rod). The optimal concentrations of wetting agents were determined optically by observation of the quality of formed coatings after the drying process (on a hot plate at 60 °C for 10 min).

The main requirements for the structure of coatings are the lack of holes and cracks. The optimal concentrations were found to be 0.025%, 0.05%, and 0.025% for Ni-Ag NPs with average sizes 100, 220, and 420 nm, respectively, and 0.35% for the polydisperse mixture (Appendix A, Appendix A). After coating and drying, the obtained films were sintered to remove the insulating layer of stabilizer from the surface of Ni-Ag NPs and to form stable joints between NPs. The sintering was performed by heating at temperatures in the range of 250–350 °C for 15 to 90 min. The electrical conductivities of obtained films were calculated from the experimentally measured resistivities (Appendix A, Appendix A) The thicknesses of all coatings, composed of Ni-Ag NPs with average sizes 100, 220, and 420 nm and the mixture of them after the sintering process were similar (~2 µm, Appendix A, Appendix A). The obtained results indicated that for all types of coatings based on Ni-Ag NPs, the optimal sintering temperature which enables obtaining the best conductivity was about 300 °C for 30 min heating (Figure 9). At these conditions, the values of conductivity of metallic films deposited with the use of monodisperse Ni-Ag NPs with average sizes 100, 220, and 420 nm were found to be 5.43·10^6^, 6.86·10^6^, and 4.86·10^6^ S/m, respectively, and in the case of model polydisperse NPs the conductivity increases to 7.87·10^6^ S/m. These values are as high as 38, 48, 34, and 55% of bulk Ni conductivity, respectively. Significant improvement in conductivity of films based on polydisperse NPs is a result of the better packing when smaller particles fill the voids between larger particles thus providing their tight packing on the surface and enabling an effective merging during sintering.

Since the highest conductivity was obtained with a coating formed by polydisperse dispersion, the optimization of its composition was performed. Figure 10 presents the conductivity (as a percent of the conductivity of bulk Ni) of metallic films after sintering at optimal conditions as a function of the weight ratios of individual NPs, which form polydisperse dispersion. The highest increase of conductivity was observed for dispersion with a 1.0:1.5:0.5 weight ratio of NPs with average sizes 110, 220, and 420 nm. The calculated values of resistivity (by multiplying the thickness ~2 µm by the value of sheet resistance 40 ± 5 mΩ/□) and conductivity (as the reciprocal value of resistivity) was 10 μΩ·cm and 9.9·10^6^ S/m that is 69% of the bulk Ni conductivity, respectively. In comparison, the sheet resistance recently reported for coatings based on Ni-Ag NPs, was only 11 mΩ/□ [30], but this value was obtained after sintering at a much higher temperature (850 °C). Since the thickness of the coating was not indicated in this report, it is difficult to compare the conductivities of the metallic coatings. The resistivity of films composed of Ni NPs after optimization of conditions of flash-light sintering was found to be 76.34 μΩ·cm [31], which was almost two times higher than obtained in the presented research. In addition, the stability of those films against the oxidation process was not analyzed.

To support our statement that significant improvement in conductivity of films based on polydisperse NPs is a result of their tighter packing compared to monodisperse NPs, the SEM analysis of sintered films was performed. Figure 11 shows SEM images of coatings composed of individual Ni-Ag NPs as well as coating composed of polydisperse NPs. The coating conditions and post-coating treatment were the same for all kinds of NPs. As clearly seen, the coating composed of the polydisperse NPs is better packed compared to those composed of monodisperse NPs. In addition, for all coatings, the changes in the shape and size of NPs are observed, but in the case of polydisperse NPs better coalescence and packing are obvious. Actually, NPs are well merged and form continuous films, which ensure their highest conductivity among the other studied coatings.

## 4. Conclusions

There is no simple answer to the question “is it rational to put so much effort into obtaining a perfect monodisperse system”? Generally, it depends on the intended application, e.g., for medical applications where the interaction of NPs with living cells/tissues/organs strongly depends on size it is crucial, while in applications like the one presented in this paper, where metal NPs were used to form compact coatings with numerous contact points between them to provide high conductivity, polydisperse NPs can offer better results than monodisperse ones. One more challenge while using metal NPs is their fast oxidation under ambient conditions. Therefore, the nickel-silver core-shell NPs, in which Ag shell provides stability of Ni core to oxidation, were synthesized, and such NPs were stable for at least 6 months both in dispersion and in the coating as well.

It was also clearly demonstrated that the polydisperse composition of metal NPs being used for conductive coatings provides better conductivity as compared with original monodisperse NPs used for ink formulation. The highest conductivity of the coatings based on polydisperse NPs was found to be 69% of that for bulk Ni.

We believe that such ink coatings composed mainly of low-cost non-noble metals with high electrical conductivity can have great potential applications in the printed electronics industry and will contribute to developing an efficient and low-cost manufacturing route for electronic devices.

## Figures and Tables

**Figure 1 materials-14-02304-f001:**
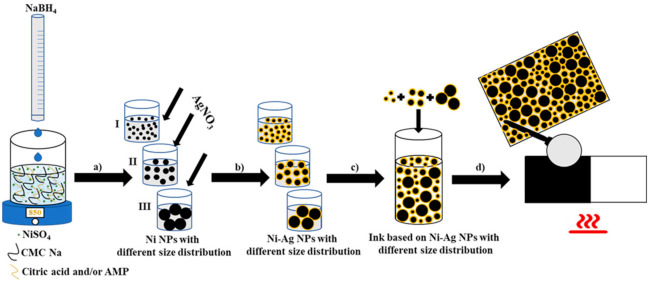
Scheme of the preparation of conductive coatings based on Ni-Ag NPs: (**a**) optimization of the process of synthesis of Ni NPs with different size distributions; (**b**) formation of Ni-Ag NPs by transmetalation method; (**c**) preparation of ink containing Ni-Ag NPs with various sizes; (**d**) fabrication of conductive metallic coating.

**Figure 2 materials-14-02304-f002:**
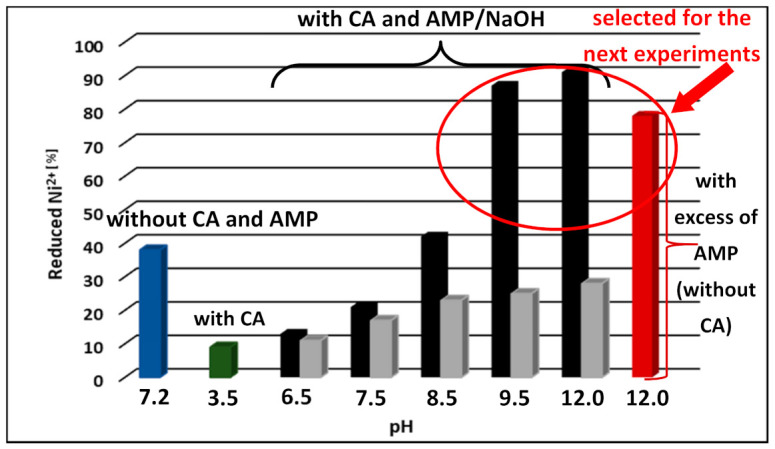
The dependence of the efficiency of Ni^2+^ reduction on the conditions of the synthesis process obtained from the EDXRF data: without complexing agents (blue bar); with citric acid (green bar); with complexing agents (black bars), with CA (citric acid) and NaOH (grey bars); with an excess of AMP (amino methyl propanol) (red bar).

**Figure 3 materials-14-02304-f003:**
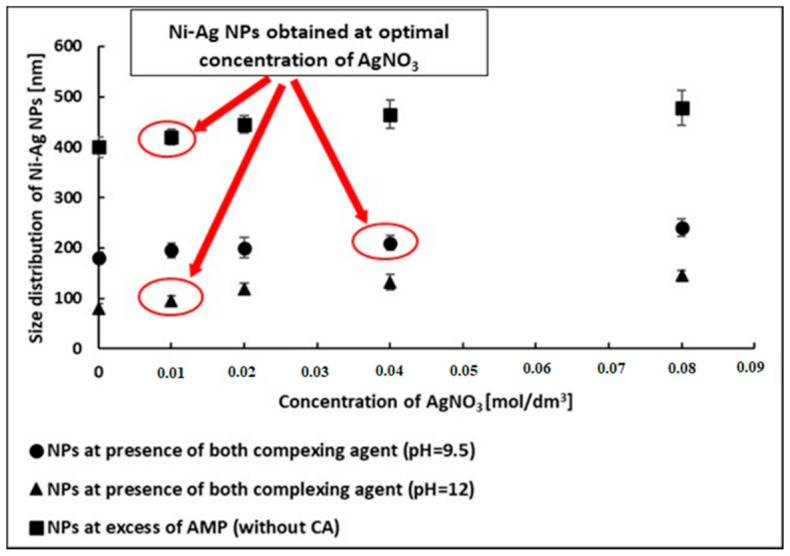
The dependence of the size of Ni-Ag NPs on the concentration of AgNO_3_ in the reaction mixture. Ni-Ag NPs obtained at the optimal concentration of AgNO_3_ are marked by red circles.

**Figure 4 materials-14-02304-f004:**
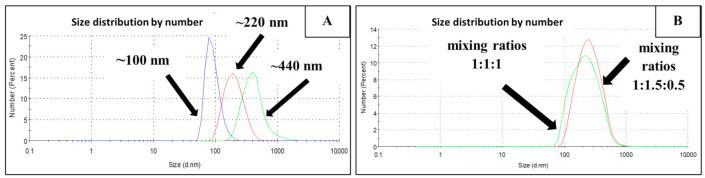
(**A**) The size distributions of the individual dispersions and (**B**) polydisperse dispersions of Ni-Ag NPs as mixtures of Ni-Ag NPs with ratio 1:1:1 and 1:1.5:0.5.

**Figure 5 materials-14-02304-f005:**
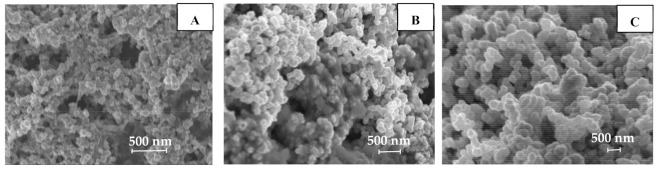
Scanning Electron Microscopy (SEM) images of Ni-Ag NPs with the size distribution of about 100 (**A**); 220 (**B**) and 420 nm (**C**).

**Figure 6 materials-14-02304-f006:**
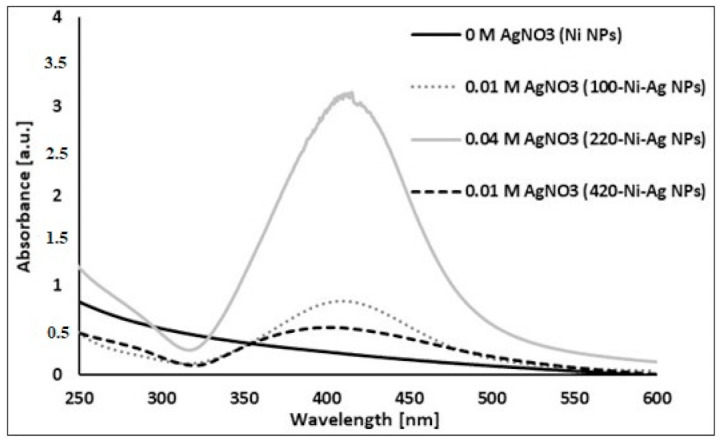
Absorbance spectra of Ni NPs (black curve) and Ni-Ag NPs at various concentrations of AgNO_3_.

**Figure 7 materials-14-02304-f007:**
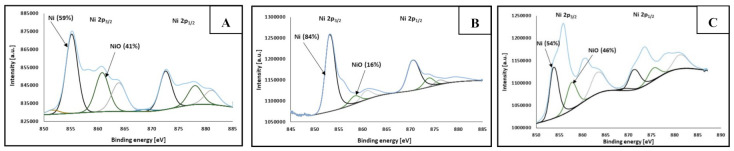
XPS spectra of Ni-Ag NPs with different sizes—analysis of nickel: (**A**) 100 nm, (**B**) 220 nm, (**C**) 420 nm (metallic Ni—black curves, NiO—green curves, satellite—grey curves).

**Figure 8 materials-14-02304-f008:**
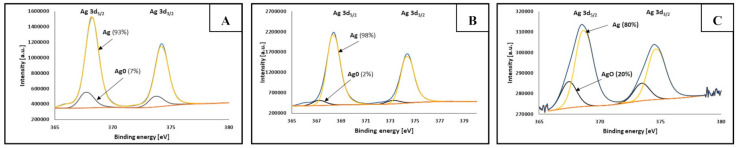
XPS spectra of Ni-Ag NPs with different sizes—analysis of silver: (**A**) 100 nm; (**B**) 220 nm; (**C**) 420 nm (spectra bands: metallic Ag—yellow curve, AgO—black curve).

**Figure 9 materials-14-02304-f009:**
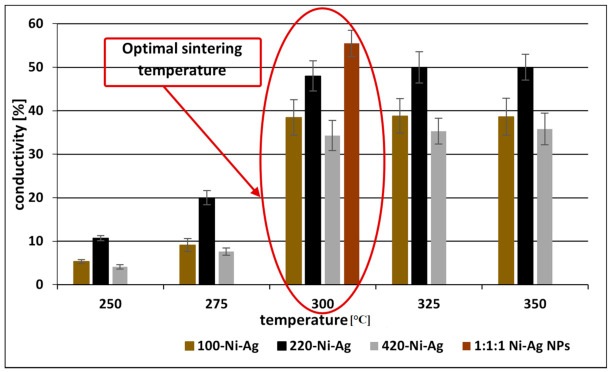
The dependence of the conductivity (in comparison to bulk nickel) of metallic films based on Ni-Ag NPs with different average sizes and their mixture (at 1:1:1 weight ratio) on the temperature of sintering (30 min).

**Figure 10 materials-14-02304-f010:**
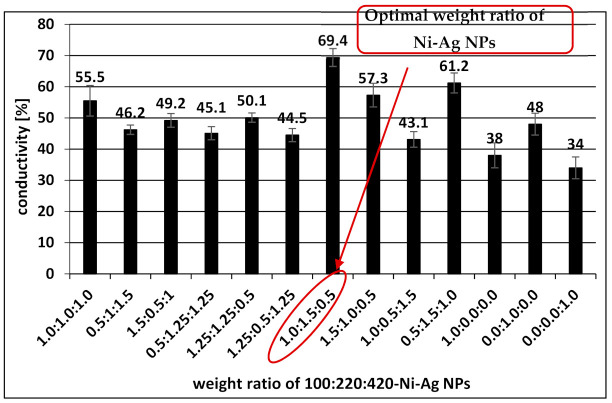
The dependence of the conductivity compared to the bulk nickel for metallic films formed with a mixture of Ni-Ag NPs with different size distributions on their weight ratio after sintering at optimal conditions (300 °C, 30 min).

**Figure 11 materials-14-02304-f011:**
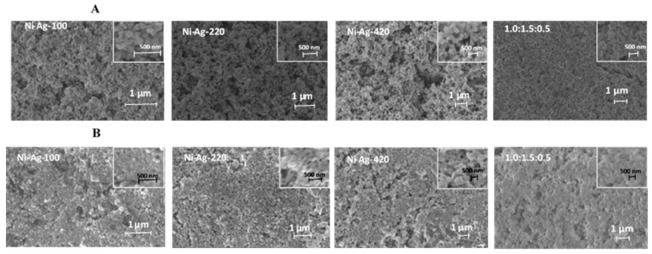
SEM images of metallic coatings formed by individual Ni-Ag NPs with different average sizes and by their mixture at an optimal weight ratio (1.0:1.5:0.5): (**A**) after drying at 60 °C for 10 min and (**B**) after sintering at 300 °C for 30 min.

**Table 1 materials-14-02304-t001:** Dependence of average size and polydispersity index for the dispersion of Ni NPs on synthesis conditions.

Synthesis Conditions	Average Size [nm]	Polydispersity Index (PDI)
AMP + CA at pH 9.5	180 ± 15	0.152 ± 0.033
AMP + CA at pH 12	80 ± 10	0.093 ± 0.021
AMP at pH 12	400 ± 25	0.211 ± 0.029

**Table 2 materials-14-02304-t002:** Chemical composition of synthesized Ni-Ag NPs according to XPS.

Average Size [nm]	Atomic Percentage [%]			
	Metallic Nickel	Nickel Oxide (NiO)	Metallic Silver	Silver Oxide
100	59	41	93	7
220	84	16	98	2
420	54	46	80	20

## Data Availability

Not applicable.

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
