# Peer review of "Polydispersity vs. Monodispersity. How the Properties of Ni-Ag Core-Shell Nanoparticles Affect the Conductivity of Ink Coatings"

_materials, 2021, doi:10.3390/ma14092304_

Round 1
Reviewer 1 Report
The authors report the effect of size distribution on the conductivity of sintered Ni-Ag films obtained from mixed inks with various sized NPs (100, 220, and 420 nm). The authors found the optimized ratio of these mixed NPs (100, 220, and 420 nm of 1:1.05:0.5) to produce the high conductive film with 1.6 times higher than that of bulk Ni. The most concept of this study is almost the same as the authors' previous one (Ref. 9). I am not sure what is substantively new in this manuscript. In previous work, the author utilized the mixed inks with the two-size distribution.
In contrast, they used three-size distribution for mixed inks in this study. The authors are encouraged to emphasize the advanced points of this study using the three-size distribution. Besides, some conclusions are not supported by experimental results. Other comments are described below.
- The preparation conditions of Ni-Ag NPs with various sizes and the inks are not clear. The reader cannot follow the preparation due to the lack of information. The author should add them in more detail as supporting information.
- What is the molar ratio of the AMP-Ni-CA complex? (i.e. AMP:Ni:CA = x:y:z)
- The author should add XPS data showing Ni(0), NiO, Ag(0), and Ag2O. The supported data of Table 2 is not shown. In general, the assignment of oxide Ni is very complex in the XPS spectra.
Thus, it is not easy to obtain the chemical composition shown in Table 2.
- The long-term oxidation stability for six months is a significant result. Still, the author does not show the XPS data.
- The author should add DLS data of Ni NPs to estimate the Ag shell thickness of Ni-Ag NPs.
- XRD data are helpful to examine the crystal structure and the alloying or not.
- The high-resolution SEM images of sintered Ni-Ag NPs from mixed ink are needed to show the sintering morphology, as shown in Fig. 5.
Reviewer 2 Report
In this article, the authors present an intriguing research on the effects of nanoparticle size and composition on the morphology and conductivity of the as-prepared coatings formed by Ni-Ag core-shell nanoparticles. The introduction is sufficient, and the results are clearly presented. However, the authors may need to address the following concerns:
- The XPS spectra of the Ni-Ag samples are missing, the authors also need to recalculate the atomic percentage in Table 2, since the sum percentages for each size should be 100%.
- XRD spectra of the as-prepared Ni-Ag nanoparticles with different sizes are needed to evaluate the effect of crystallinity on the conductivity.
- The control experiment for Ni nanoparticles without Ag coating is missing.
- The resolution of the figures and graphs in the article needs to be improved.
Round 2
Reviewer 1 Report
The authors have addressed the issues in a due way. I suggest its acceptance.
Reviewer 2 Report
I suggest the authors put Figure 1-6 in the supporting information in the main text. They also need to provide a detail analysis on the XPS spectra, i.e., identify different chemical bonds and interactions of the as-prepared Ni-Ag NPs.
